# Green Synthesis of Gold and Silver Nanoparticles by Using *Amorphophallus paeoniifolius* Tuber Extract and Evaluation of Their Antibacterial Activity

**DOI:** 10.3390/molecules25204773

**Published:** 2020-10-18

**Authors:** S. M. Abu Nayem, Nasrin Sultana, Md. Aminul Haque, Billal Miah, Md. Mahmodul Hasan, Tamanna Islam, Md. Mahedi Hasan, Abdul Awal, Jamal Uddin, Md. Abdul Aziz, A. J. Saleh Ahammad

**Affiliations:** 1Department of Chemistry, Jagannath University, Dhaka 1100, Bangladesh; nayemjnu96@gmail.com (S.M.A.N.); nasrinjnu98@gmail.com (N.S.); amin2k50@gmail.com (M.A.H.); billaljnu8@gmail.com (B.M.); mahmodjnu@gmail.com (M.M.H.); islamtamanna1992@gmail.com (T.I.); mdhasanmahedi83@gmail.com (M.M.H.); aawal4513@gmail.com (A.A.); 2Center for Nanotechnology, Department of Natural Sciences, Coppin State University, 2500 W. North Ave, Baltimore, MD 21216, USA; 3Center of Research Excellence in Nanotechnology, King Fahd University of Petroleum and Minerals, Dhahran 31261, Saudi Arabia

**Keywords:** green synthesis, nanoparticles, AuNPs, AgNPs, antibacterial activity

## Abstract

In this report, we discussed rapid, facile one-pot green synthesis of gold and silver nanoparticles (AuNPs and AgNPs) by using tuber extract of *Amorphophallus paeoniifolius*, and evaluated their antibacterial activity. AuNPs and AgNPs were synthesized by mixing their respective precursors (AgNO_3_ and HAuCl_4_) with tuber extract of *Amorphophallus paeoniifolius* as the bio-reducing agent. Characterization of AuNPs and AgNPs were confirmed by applying UV-vis spectroscopy, field-emission scanning electron microscopy (FESEM), X-ray diffraction (XRD) analysis, Fourier transform infrared spectroscopy (FTIR), and energy dispersive X-ray spectroscopy (EDS). From UV-vis characterization, surface plasmon resonance spectra were found at 530 nm for AuNPs and 446 nm for AgNPs. XRD data confirmed that both synthesized nanoparticles were face-centered cubic in crystalline nature, and the average crystallite sizes for the assign peaks were 13.3 nm for AuNPs and 22.48 nm for AgNPs. FTIR data evaluated the characteristic peaks of different phytochemical components of tuber extract, which acted as the reducing agent, and possibly as stabilizing agents. The antibacterial activity of synthesized AuNPs and AgNPs were examined in Muller Hinton agar, against two Gram-positive and four Gram-negative bacteria through the disc diffusion method. AuNPs did not show any inhibitory effect, while AgNPs showed good inhibitory effect against both Gram-positive and Gram-negative bacteria.

## 1. Introduction

In this era of modern science and technology, nanoparticles and nanomaterials are widely used for various purposes. Nanoparticles have numerous applications in the field of electronics, photochemicals, biomedicine, catalysis, and display devices due to their exceptional properties, such as high surface area, quantum confinement properties, variable size, and shape-dependent properties [1,2,3]. In recent times, inorganic nanoparticles, mainly noble metal nanoparticles, have gained considerable interest due to their biocompatibility and higher surface to volume ratio compared to their bulk counterpart. Among these noble metal nanoparticles, AuNPs and AgNPs are extensively used in different fields, such as in biosensing applications, anticancer, anti-fungal and antioxidant agents, and medical treatment [4,5,6]. Besides these applications, AuNPs are used in dye degradation, water hygiene management, and additionally showed anti-HIV activity [7,8,9]. AgNPs are widely used in the field of medicine as an anti-bacterial agent. In particular, they exhibit inhibitory effect against both Gram-positive and Gram-negative bacteria, hence the silver base product has gained commercial demand in the field of food coating, contact lenses, surgical masks, and in wound dressings. The use of Ag-based products is approved by the global regulatory committee. Besides these uses, AgNPs are used in catalysis, optics, and as anti-biofilm agent [6,10].

Increasing demand for AuNPs and AgNPs in different sectors has led to large scale production of these nanoparticles. These nanoparticles are usually produced through conventional methods, such as physical [11] and chemical [12]. These methods include laser ablation, sol-gel, pyrolysis, chemical, and reduction methods. Most of this process makes the whole process costly and hazardous to the environment and public health. Preparation of nanoparticles through physical methods needs high temperature, pressure, and costly instrumental setup, while chemical methods often require hazardous chemical substances and additional reagents, as a reducer and stabilizer. Moreover, synthesized nanoparticles adsorb hazardous substances on their surface, making them inconvenient in biomedical uses [13,14].

To avoid large-scale production costs and the use of hazardous chemicals, researchers have started transitioning towards green synthesis methods to produce nanoparticles. Green synthesis of AuNPs and AgNPs consists of using plant, fungi, virus, and bacteria as a source of reducing and stabilizing agent [15,16,17,18]. From the literature survey, it is found that nanoparticles synthesized with plant-extracts show more biocompatibility. The plant extracts contain different types of phytochemical components, namely: alkaloids, steroids, carbohydrates, proteins, etc. [19]. The presence of these phytochemicals is responsible for the reduction of metal ions into metal nanoparticles and further colloidal stabilization [20]. Different plant extracts have been used in recent times to synthesise AuNPs and AgNPs, such as *Zingiber officinale* [21], *Ficuscarica* [22] and *Polyalthia longifolia* [23]. Aside from these, there are many more reports where AuNPs and AgNPs were synthesized from plant extracts.

*Amorphophallus paeoniifolius* is an herbaceous, perennial C3 crop that is categorized under the family Araceae; it has yellow flesh and is a rich source of starch [24,25]. This plant is also known as elephant foot yam and has the local name, “Olkochu”. This plant is widely available in Bangladesh at very low cost. This medicinal plant contains different phytochemical components such as steroids, flavonoids, carbohydrates, tannins, saponins and proteins [26]. Due to the presence of different bioactive molecules, this plant has many uses in medicinal purposes such as cough, vomiting, bronchitis, asthma, and anorexia [14,27]. Besides these medicinal uses, the above-mentioned phytochemicals in this plant can be used for synthesizing metal nanoparticles through reducing metal ions [10]. A recent work reported synthesis of AgNPs by using leaf extract of *Amorphophallus paeoniifolius*, and their potential antibacterial activity [14].

In this manuscript, we reported cost-effective, simple, rapid, and eco-friendly synthesis of Au and Ag NPs at room temperature. Tuber extract (without bark) of *Amorphophallus paeoniifolius* was used as the source of reducing and stabilizing agents. Using this tuber extract as a reducing and stabilizing agent, instead of toxic chemicals, satisfies the main object of green chemistry [28]. The formation of AuNPs and AgNPs was confirmed by using the UV-vis spectrophotometric technique. Their crystalline nature was identified by using the X-ray diffraction method (XRD). The surface morphology of these nanoparticles was evaluated by using field-emission scanning electron microscopy (FESEM). Elemental compositions present in the synthesized nanoparticles were confirmed by energy dispersive spectra (EDS). The functional groups present in the tuber extract and in synthesized nanoparticles were evaluated using Fourier transform infrared spectroscopy (FTIR). The antibacterial activity of these synthesized AuNPs and AgNPs was evaluated on Muller Hinton agar using the disc diffusion method, against both Gram-positive and Gram-negative bacteria.

## 2. Results and Discussion

Water-soluble components present in tuber extract of *Amorphophallus paeoniifolius* were responsible for the reduction of gold trivalent and silver monovalent ions into AuNPs and AgNPs [14]. Preliminary evidence of nanoparticle formation was confirmed from color change. The mixture of tuber extract and aqueous solution of HAuCl_4_ turned yellowish to pink while the mixture of tuber extract and aqueous solution of AgNO_3_ turned yellowish to reddish-brown. The colors of the synthesized AuNPs and AgNPs are shown inset Figure 1. Similar changes in color were reported for the synthesis of AuNPs and AgNPs in previous studies [29,30].

### 2.1. UV-Vis Spectroscopy

UV-vis spectroscopy is a technique that can confirm the formation of metal nanoparticles in aqueous solution [4]. The UV-vis spectrum of tuber-mediated AuNPs colloidal solution is shown in Figure 1A. The characteristic surface plasmon resonance (SPR) absorption band of AuNPs, appearing at a wavelength of 530 nm, indicated the formation of AuNPs [31,32]. Figure 1B displayed the UV-vis spectrum of AgNPs colloidal solution. A broad SPR band arose at 446 nm, indicating the formation of AgNPs [33,34]. The SPR of AuNPs and AgNPs was attributed to the interaction between free electrons on the metal surface and incident light [35]. 

### 2.2. X-ray Diffraction Analysis

The crystalline nature of synthesized AuNPs and AgNPs was determined by using X-ray diffraction (XRD) analysis [36]. Figure 2A,B show the XRD data of synthesized AuNPs and AgNPs. The 2θ range for both dried AuNPs and AgNPs was 10–80°. For AuNPs, sharp intense peaks observed at 38.24°, 44.42°, 64.86°, and 77.67° correspond to the Braggs plane (111), (200), (210), and (310), which confirmed that the synthesized AuNPs had a face-centered cubic structure. [8] The average crystallite size for all assigned peaks of the synthesized AuNPs was obtained at about 13.3 nm by using the Debye Scherrer formula D = 0.9 λ/β Cosθ, where θ is the Bragg’s angle, λ is the X-ray wavelength (1.5406 A°), β is the full width at half maximum (FWHM) and D is the average crystallite size [37]. From the crystallite size, dislocation density δ was calculated as 5.65 × 10^−3^ (nm)^−2^ for AuNPs, based on the following relation δ = 1/D^2^. [38] For AgNPs, the peak obtained at 2θ values at 27.96°, 32.34°, 38.19°, 44.47°, 46.35°, 57.98°, 64.48° and 77.47° correspond to Braggs reflection (210), (122), (111), (200), (231), (241), (220) and (311) [39,40]. This indicated that synthesized AgNPs were faced center cubic in the crystalline nature. The average crystallite size for all assigned peaks was calculated to be about 22.48 nm by using the Debye Scherrer equation. The dislocation density δ was calculated as 1.98 × 10^−3^ (nm)^−2^ using the formula δ = 1/D^2^. Table 1 shows the value of crystallite size of each assigned peak of AuNPs and AgNPs and the corresponding average crystallite size values. The unassigned peaks on the surface of both AuNPs and AgNPs in XRD analysis might come from the crystallization of bioorganic molecules of tuber extract [10].

### 2.3. FESEM and EDS Analysis

FESEM imaging is an analytical technique that can be used for determining the average size and shape of nanoparticles in the test sample [5,6]. EDS analysis was used for determining the different elements present in the sample. [6] Figure 3A,B show the low magnification FESEM images of tuber-mediated synthesized AuNPs. Figure 3D,E show the low magnification FESEM images of tuber-mediated synthesized AgNPs. The low magnification FESEM images show slight aggregation for both AuNPs and AgNPs. However, the shapes of AuNPs and AgNPs cannot be discerned from these low magnification images. The SPR peaks observed from UV-vis indicated that Au and Ag NPs with low aggregation were synthesized through the proposed method. Hence, it is likely that the metal NPs aggregated during sample preparation for SEM analysis. The high magnification FESEM images (Figure 3C,F) for AuNPs and AgNPs show mostly spherical particles along with a few polygonal shaped particles. The presence of elemental gold was confirmed from the strong absorption peak, shown in Figure 3G, using EDS analysis at 2.15 keV of tuber-extract synthesized AuNPs [8,36]. A strong absorption peak appeared at 3.0 keV in EDS analysis and is displayed in Figure 3H, indicating the presence of elemental silver [41]. The peaks found in both AuNPs and AgNPs EDS in the range of 0–0.5 keV indicated the presence of carbon and oxygen [10]. The peak of Sn and Si appeared in the EDS spectrum due to the FTO coated glass substrate that was used for sample preparation. Other absorption peaks in the EDS of both AuNPs and AgNPs might come from impurities in tuber extract or from the FESEM grid.

### 2.4. FTIR Analysis

FTIR spectroscopy was employed to deduce the possible biomolecules in the tuber extract that were responsible for the reduction of Au^3+^ to Au^0^ and Ag^+^ to Ag^0^ [8,14]. FTIR spectroscopy analysis also revealed the possible biomolecules and functional groups responsible for capping or stabilizing of the synthesized AuNPs and AgNPs. FTIR spectra of *Amorphophallus paeoniifolius* tuber extract, AuNPs, and AgNPs were expressed in Figure 4. Taking the spectrum of tuber extract as control, the involvement of different functional groups of *Amorphophallus paeoniifolius* tuber extract, in reducing and stabilizing process of nanoparticles synthesis, was evaluated. Absorbance bands at 3419, 2934, 1634, 1336, 1408, 1080, and 775 cm^−1^ were observed in the spectrum of *Amorphophallus paeoniifolius* tuber extract. A broad band at 3419 cm^−1^ was due to the O-H stretching vibration of alcohol or phenolic compounds. A weak peak at 2934 cm^−1^ was due to the asymmetric stretching vibration of the C-H bond of alkane [42]. The peaks at 1634 cm^−1^ and 1336 cm^−1^ could be assigned due to the characteristic band of –NH_2_ groups containing amino acids of protein and C=O groups of flavonoids and tannins [14,23]. Due to the angular deformation of the C-H bond, a sharp intense peak was obtained at 1408 cm^−1^ [24]. For ether linkages and C-O or C-O-C groups, a peak appeared at 1080 cm^−1^ and 775 cm^−1^, which could be attributed to the deformation vibration of C-H bonds of the phenolic ring [14,41]. A high degree of similarity was observed in the FTIR spectrum of AuNPs and AgNPs in comparison to the tuber extract (control) spectrum. For AuNPs, the peak appeared at 3429, 2926, 1627, 1408, 1097 and 767 cm^−1^. For AgNPs, the peak position was 3435, 2927, 1626, 1382, 1074 and 655 cm^−1^. The main absorbance band of AuNPs and AgNPs was shifted and the intensity also reduced in comparison with the control spectrum. This shifting and reduction in peak intensity of main absorbance band of AuNPs and AgNPs revealed that biomolecules present in tuber extract were responsible for the reduction in gold and silver salt. [8,14,41] However, the drastic reduction in the FTIR peaks compared to the control is also indicative of low stabilization of the synthesized NPs [4,10]. 

### 2.5. Anti-Bacterial Analysis with AuNPs and AgNPs

The potent inhibitory effects of tuber-mediated synthesized AuNPs and AgNPs on six pathogenic bacteria were investigated on Muller Hinton agar using the disc diffusion method. AuNPs did not show any inhibitory effect due to the nontoxic nature of Au [30], while AgNPs showed good antibacterial activity against both Gram-positive and Gram-negative human pathogenic bacteria. The photographic image of antibacterial activity exhibited by AgNPs against all six pathogenic bacteria is shown in Figure 5. The zone of inhibition (ZOI) was measured for AgNPs, against all these six micro-organisms, and is shown in Table 2, with comparisons to standard drug ciprofloxacin. It was noticed that Gram-negative bacteria show higher ZOI than Gram-positive bacteria. The exact mechanisms of the inhibitory action of AgNPs on pathogenic microorganisms are not very well known [43]. However, several mechanisms of the inhibitory effect of AgNPs on different pathogenic bacteria are reported. The positive charge of AgNPs and the negative charge of the bacteria cell wall interact, leading to cell death [6]. There are three different modes of the anti-bacterial effect of AgNPs on bacterial cell walls according to the literature review: (i) releases of Ag^+^ ion, (ii) release of reactive oxygen species (ROS), and (iii) particle size [14].

Firstly, previous reports have indicated that the colloidal solution of AgNPs contains both adsorbed and free Ag^+^ along with solid AgNPs [14,36]. These AgNPs gradually release Ag^+^ ion, which can interact with the negatively charged bacterial cell membrane. These interactions may lead to disruption of the integrity function of bacteria, leading to cell death. Ag^+^ ions can penetrate the cell membrane of bacteria; due to the soft acidic nature of silver, this ion can interact with the sulphur and phosphorus bases of DNA, which inactive the replication ability of DNA by deforming the structure, resulting in cell death. Besides this, the silver ion can interact with the ribosome, resulting in the suppression or expression of proteins and enzymes, thus taking part in the essential metabolism of bacteria [14,30,44].

Secondly, the research has indicated that different types of ROS, such as superoxide anion and radical species, may form through the binding of AgNPs and the cell membrane of bacteria and mesosoms. This production of ROS may cause a disruption in the function of mesosoms, resulting in cell death [5,30].

Thirdly, small size nanoparticles have higher surface area, hence interaction with bacteria occurred to a greater degree, resulting in intervening bactericidal function [45]. From the literature review, it was found that small size nanoparticles have a smaller number of the unit cells and most of the unit cells appeared on the surface of nanoparticles. Hence all of these unit cells are involved in the bacteria cell distortion process by the above three mentioned factors [14]. The presence of small size AgNPs is greatly absorbed by the surface of bacterial cell wall resulting in a greater amount of interaction with bacteria [5]. This greater amount of interaction of nanoparticles is evaluated through higher ZOI.

The reason for higher ZOI of Gram-negative bacteria than Gram-positive might be the following: the cell wall of Gram-positive bacteria contains teichoic acid, while the outer membrane of Gram-negative bacteria contains lipopolysaccharide and phospholipids. Due to this structural difference, their cell wall imparts different negative charge. Besides this, the Gram-negative bacterial cell wall contains more negative charge than Gram-positive bacteria, and the Gram-positive bacterial cell wall is formed by a more rigid and thick peptidoglycan layer than that of the Gram-negative bacteria. These different properties of two different types of bacteria indicate that penetrating and electrostatic attraction of AgNPs is greater with Gram-negative bacteria than with Gram-positive bacteria, resulting in higher ZOI of Gram-negative bacteria than Gram-positive bacteria [14,46].

### 2.6. Effect of Sunlight in AgNPs Synthesis

For AgNPs synthesis, the mixture of aqueous AgNO_3_ solution and tuber extract was exposed to sunlight; within a few minutes (about 2–3 min), the color of the solution changed from yellowish to reddish-brown in dark condition. The same mixture of aqueous AgNO_3_ solution and tuber extract was subjected to sunlight for about one hour, and no color change was observed. The individual solution of AgNO_3_ and tuber extract also exposed to sunlight for about 10 min. The color of the solution remained unchanged. This indicated that sunlight played a vital role in the reduction of Ag^+^ ion into Ag^0^ formation. The effects of sunlight in AgNPs synthesis are still unknown. However, in some studies, the mechanism of sunlight in bio reduction of AgNPs is reported. We propose here a probable mechanistic effect of sunlight in AgNPs based on the literature review; it may be revealed that silver solution interacts with phytochemicals components (mainly oxygen-containing biomolecules) and forms Ag_2_O. Due to the semiconductor nature and narrow band gap (about 2.25 eV) of Ag_2_O, when sunlight interacts with the reaction mixture, electrons may jump to the conduction band of Ag_2_O, creating a hole in the valance band. These electrons may attach to the Ag^+^ ion and rapidly convert the Ag^+^ ion into Ag^0^ [46]. In another report, the effect of sunlight is described as a photosensitizer of biomolecules. When silver nitrate dissolves in water, it dissociates into the Ag^+^ ion. Then, the biomolecules of plant extract may combine with the Ag^+^ ion and form a complex ion. The exposures of sunlight into the reaction mixture may excite the oxygen-containing biomolecules (mainly –O-H groups) and release electrons through the de-bonding of –O-H. Then this releasing electron may reduce the Ag^+^ ion into Ag^0^ and form AgNPs suspension [47].

### 2.7. Probable Mechanism of AuNPs and AgNPs Formation

From FTIR and sunlight effect analysis, it would be concluded that the phytochemical components are responsible for the reduction of AuNPs and AgNPs and also the stabilization of these nanoparticles. For AgNPs, sunlight plays a vital role in the reduction reaction. *Amorphophallus paeoniifolius* tuber extract contains different phytochemicals, namely steroids, flavonoids, carbohydrates, tannins, saponins and proteins. From the FTIR study, it was seen that different phytochemicals were present on the surface of synthesized AuNPs and AgNPs, and particularly, that phytochemicals were responsible for the reduction and stabilization of synthesized nanoparticles. There have been several proposed mechanisms and different pathways are reported by using plant extract as a source of reducing and stabilizing agents. In the literature, it is reported that polyphenols of flavonoids and tannin provide the electrons for the reduction of Ag^+^ to AgNPs [47]. Flavonoid is a strong antioxidant biomolecule, which contains phenolic groups and shows keto-enol tautomerism. This antioxidant nature of flavonoid leads to the provision of electrons or reactive hydrogen to the Au^3+^ and Ag^+^ ion through the de-bonding of the –O-H bond of its enol form [42,47]. In the process of AgNPs synthesis, photon energy from sunlight is absorbed by the flavonoids. This energy accelerates the de-bonding of the –O-H bond of flavonoids, resulting in the rapid release of electrons or reactive hydrogen to reduce the Ag^+^ ion to form Ag^0^ [23,42]. From the above study, the probable mechanism of both AuNPs’ and AgNPs’ synthesis process would be proposed based on the following.

Firstly, the reduction of metal ions (Au^3+^ and Ag^+^) into nanoparticles:

From FTIR spectra analysis, it was noticed that the wave number, due to hydroxyl groups shifting towards higher wave numbers for both nanoparticles (for tuber extract, 3419 cm^−1^; AuNPs 3429 cm^−1^ and AgNPs 3435 cm^−1^). From FTIR study, it was also observed that the wave number, due to C=O groups, shifted towards a lower wave number (for tuber extract, 1634 cm^−1^; AuNPs 1628 cm^−1^ and AgNPs 1626 cm^−1^). *Amorphophallus paeoniifolius* tuber extract contains flavonoids; therefore, it can be stated that the polyphenolic groups of flavonoids were present on the surface of both nanoparticles and the oxidation of hydroxyl groups of flavonoids to ketonic groups occurred [48]. That result was helpful to postulate the probable mechanism of nanoparticles synthesis. The hydroxyl groups of flavonoids oxidized to keto form, which indicated the release of reactive hydrogen during that conversion. From the literature survey, it is obtained that the bond dissociation energy of hydroxyl groups of the catechol moiety of flavonoid (quercetin) is less than other hydroxyl groups of aromatic rings [49]. Therefore, flavonoids (quercetin) of tuber extract might react with the metal ion through the hydroxyl groups of the catechol moiety, and reduce the metal ions into nanoparticles. In the process of AgNPs synthesis, flavonoids (quercetin) might act as a photosensitizer. Quercetin probably absorbed the photon energy from sunlight and accelerated the debonding of hydroxyl groups of the catechol moiety, resulting in the rapid reduction of the Ag^+^ ion into AgNPs.

Secondly, the stability of synthesized nanoparticles:

The enol forms of quercetin could be oxidized to keto form with the reduction of the metal ion into nanoparticles. That keto group might be attached on the surface of nanoparticles to prevent aggregation of nanoparticles. Therefore, stability of synthesized nanoparticles is achieved through the chelation of nanoparticles and keto groups of the catechol moiety of quercetin. The proposed mechanism for AgNPs synthesis is shown in Scheme 1.

## 3. Experimental

### 3.1. Materials

All the chemicals and reagents were of analytical grade. HAuCl_4_ (chloroauric acid) and AgNO_3_ (silver nitrate) were purchased from Sigma-Aldrich (Darmstadt, Germany). These chemicals were used without any further purification. Distilled water was used for the whole experiment. The glassware and falcon tube were primarily washed with tap water, then rinsed with distilled water and air-dried before use in the experimental section. Microbiological media were purchased from HI Media Laboratories, Pvt. Ltd., Mumbai, India. *Amorphophallus paeoniifolius* plant was collected from local villages of the city of Jashore, Bangladesh.

### 3.2. Preparation of the Tuber Extract

*Amorphophallus paeoniifolius* plant tuber was washed through tap water for removal of mud, dirt, and foreign materials. Then, the tuber was cut into segments and all the bark was removed from the each of the segments. The tuber segments without bark were dried in shade at room temperature. Then, the dried segment was pulverized through a domestic blender to form powder. Then, 0.1 g of that powder was taken into a 250 mL reagent bottle with 100 mL distilled water and sonicated for about 90 min for crude extract formation. Then, the prepared crude extract was filtered by cellulose nitrate filter paper to avoid cellulose in the final extract. Then, the extract was stored into a refrigerator at 4 °C for nanoparticle synthesis.

### 3.3. Synthesis of AuNPs and AgNPs

The metal nanoparticles were synthesized from the plant extract following the previously mentioned reports [10,14]. The falcon tubes used for the synthesis purpose were newly purchased and cleaned with distilled water several times before use. No strong acids/oxidizing agents were used throughout the experimental works. For the synthesis of AuNPs, 0.8 mM aqueous solution of chloroauric acid (HAuCl_4_) and the tuber extract with 1:4 volume ratios (volume of chloroauric acid solution was 100 µL and extract volume was 400 µL) were taken into the falcon tube for reduction of Au^3+^ ion; after 1 h, the formation of AuNPs was visually observed through color change from yellowish to pink. For AgNPs synthesis, an aqueous solution of 0.01 M silver nitrate (AgNO_3_) and tuber extract with 1:1 volume ratio (both volumes were 500 mL) were taken into a falcon tube and exposed to bright sunlight for reduction of Ag^+^. Within a few minutes (about 2–3 min), the formation of AgNPs was visually observed through color change from yellowish to reddish-brown.

### 3.4. Characterization of Synthesized AuNPs and AgNPs

UV-vis spectra of AuNPs and AgNPs were recorded using a SHIMADZU UV-1800 double-beam spectrophotometer (Shimadzu Corporation, Kyoto, Japan) in the range of 400–700 nm for AuNPs and 200–800 nm for AgNPs, at a resolution of 1 nm. Spectra were obtained from 3 mL of both AuNPs and AgNPs (2 times dilution) volumes in a 1 cm path length quartz cuvette. Distilled water was set as baseline correction. The X-ray Diffractometer (XRD, Ultima IV, Rigaku Corporation, Tokyo, Japan), with a Cu Kα radiation source, was used to determine the crystalline nature of dried AuNPs and AgNPs, in the range of 20 to 80° and at a scanning rate of 3°/min^−1^ at 40 kV and 40 mA, with step width of 0.02°. The surface morphologies of synthesized AuNPs and AgNPs were characterized from FESEM analysis by using a field-emission scanning electron microscope (FESEM, TESCAN LYRA 3, Tescan, Brno, Czech Republic); the elemental composition of synthesized nanoparticles was confirmed from energy-dispersive X-ray spectra (EDS) by using an Oxford Instruments Xmass detector (Oxford Instruments, Abingdon, United Kingdom) equipped with the FESEM. EDS analysis was done from specific points of FESEM images. The dual role of tuber extract as reducing and stabilizing agent of synthesized nanoparticles was confirmed by using FTIR spectra. Appropriate quantities of KBr and sample (extract and both synthesized nanoparticles in powder form) in the ratio of 100:0.1 were mixed by grinding in an agate mortar. Pellets were made with about 100 mg mixture. FTIR spectra were recorded with the FTIR 8400 S Shimadzu spectrophotometer (Shimadzu Corporation, Kyoto, Japan) in the range of 4000–400 cm^−1^. Resolution: No. of scans: 50 times.

### 3.5. Antibacterial Assay

Antibacterial activity of *Amorphophallus paeoniifolius* tuber-mediated synthesized AuNPs and AgNPs was investigated on Mueller Hinton agar using disc diffusion method, against two Gram-positive (*Bacillus subtillis, Staphylococcus aureus*) and four Gram-negative (*Pseudomonas aeruginosa, Escheirichia coli, Salmonella typhimurium, Citrobacter freundii*) bacteria, as a test microorganism. In this study, 38 g/L autoclaved Mueller Hinton agar acted as a cultured media for the antibacterial analysis. Saline water (0.8 g NaCl/100 mL) was used in this analysis as a mediator of bacteria growth. All the six cultured bacteria dissolving into saline water were lapped by using sterile cotton buds on Muller Hinton agar with six different freshly oven-dried petri plates. Besides this, sterile autoclaved filter paper discs were subjected to the Petri plates, then 25 µL of tuber-mediated synthesized nanoparticles were injected on filter paper discs. After 48 h of incubation time, at 37 °C the zone of inhibition was measured in millimeters by using a meter scale.

## 4. Conclusions

In this report, we discussed a green, rapid, one-pot synthesis of AuNPs and AgNPs, and evaluated their antibacterial activity. HAuCl_4_ and AgNO_3_ were used as a source of gold and silver, and tuber extract of *Amorphophallus paeoniifolius* was used as reducing and stabilizing agent in this synthesis pathway. UV-vis spectra showed a characteristic SPR band of 530 and 446 nm for synthesized AuNPs and AgNPs. XRD data evaluated the crystallinity of these synthesized nanoparticles; it was seen that both AuNPs and AgNPs were face-centered cubic and the average crystallite sizes for the assign peaks were 13.3 and 22.48 nm. FTIR spectra indicated the presence of phytochemicals that were responsible for the reduction and possible stabilization processes. Anti-bacterial activity of synthesized nanoparticles was analyzed by the disc diffusion method against two Gram-positive and four Gram-negative bacteria. AuNPs did not show any inhibitory effect, while AgNPs showed good inhibitory effect. Sunlight played a vital role in the synthesis of AgNPs as it photosensitizes the biomolecules present in the tuber extract. The optimization of the different parameters in this synthesis procedure needs further study. Tuber extract contains different phytochemicals, hence it is not exactly clear which biomolecules are responsible for the reduction and stabilization processes. Due to the presence of different biomolecules with different antioxidant nature, preparation of nanoparticles of specific size is difficult. These above difficulties will be removed when the exact biomolecules that are responsible for the reduction and stabilization process are identified.

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
