# Peer review of "Green Synthesis of Gold and Silver Nanoparticles by Using Amorphophallus paeoniifolius Tuber Extract and Evaluation of Their Antibacterial Activity"

_molecules, 2020, doi:10.3390/molecules25204773_

Round 1
Reviewer 1 Report
Here the authors report a “bottom-up” synthesis of both gold and silver nanoparticles (NPs) prepared in aqueous solution with a plant extract acting as the source of both reducing and stabilizing agent(s).
The work reported here seems rigorous enough for the most part, and the English writing is of quality for most of the manuscript. I suggest that the authors revise some of their statements, particularly when advancing mechanism(s) that are mostly unfounded (see comments below).
- 65-67: There are hundreds of gold or silver NPs syntheses with plant extracts reported in the literature. The authors do not need to cite them comprehensively, however they should point to the fact that there are many more than the handful cited here.
- 76-77 and 79-81: English writing needs to be corrected here.
- 78: Could the authors elaborate as to why they used the tuber, not the leaves, of this plant? Also, why this plant? Is it local?
l.94: It is standard procedure in the synthesis of metal NPs to clean the glassware with a strong acid/strong oxidizer such as HNO3 or aqua regia, to remove any trace of metal. Was any of this done, between runs or after a number of runs?
- 155 (Figure 1): How many replicates were performed for each synthesis? Also, how reproducible was each synthesis? What was the variation in SPR peak shape and maximum?
- 167, 187, 188, 191: citations errors
- 185: I disagree with the statement that the NPs were “spherical”. One could tell with a higher resolution TEM, but not with the SEM provided.
- 190: SEM is not an in situ method. As such, when we observe NP aggregation, we do not know whether it resulted from SEM sample preparation, or whether the aggregation was already present in the reaction mixture. That is different from stating that the aggregation observed comes from the SEM preparation (we do not know, although the SPR peak shape reported earlier likely indicates low aggregation).
- 203: The biomolecules observed with FTIR are present in the sample. However, we do not know whether they are adsorbed on the NP surface (if stabilizing agents) or are residues within the mixture. Also, if they acted as reducing agents, we would see a chemical change, and I am not convinced that this is the case here.
- 244-251: Although there is some consensus as to how silver NPs act as antimicrobial agents (by releasing Ag+ ions), I would write as “it is believed”, “research has indicated” etc. instead of describing it as a known fact.
- 252-254: Comment above applies to ROS here.
- 272- and l. 280-: This is a very interesting addition. Experimentally, sunlight increases the rate of nucleation and/or growth in the synthesis silver nanoparticles. The “proposed” mechanism as to how sunlight actually acts in silver NP formation is not a fully known fact, and should not be described as such.
- 291-: Mechanisms are always “proposed” and can never be 100% “proven”. Here the authors describe a mechanism without any quantitative experimental data to support it. That is not acceptable. They can “propose” a mechanism based on their qualitative observations and from what Is known in the literature, as an interesting addition. That would be fundamentally different from advancing a mostly unfounded mechanism.
Author Response
1st Reviewers' Comments:
Here the authors report a “bottom-up” synthesis of both gold and silver nanoparticles (NPs) prepared in aqueous solution with a plant extract acting as the source of both reducing and stabilizing agent(s).
The work reported here seems rigorous enough for the most part, and the English writing is of quality for most of the manuscript. I suggest that the authors revise some of their statements, particularly when advancing mechanism(s) that are mostly unfounded (see comments below).
Comment 1. 65-67: There are hundreds of gold or silver NPs syntheses with plant extracts reported in the literature. The authors do not need to cite them comprehensively; however, they should point to the fact that there are many more than the handful cited here.
Response 1. Thank you for your kind comment. We modified the line as the reviewer suggested. Please see lines: 66-67 of the revised manuscript for details.
Comment 2. 76-77 and 79-81: English writing needs to be corrected here.
Response 2. Thank you for your kind comment. We have corrected our English writing as the reviewer suggested. Please see the revised manuscript.
Comment 3. 78: Could the authors elaborate as to why they used the tuber, not the leaves, of this plant? Also, why this plant? Is it local?
Response 3. Thank you for your kind comment. We chose the tuber portion of the plant for nanoparticles synthesis as leaf extract based AgNPs synthesis has recently been reported. Please see the reference 14 from the revised manuscript.
We are sorry for any confusion caused by our presentation. Amorphophallus paeoniifolius is a locally grown plant in Bangladesh. It is a cheap plant and can grow anywhere with little care. We have discussed about this in more details in the revised manuscript. Please see in line 68-77 of the revised manuscript.
Comment 4. It is standard procedure in the synthesis of metal NPs to clean the glassware with a strong acid/strong oxidizer such as HNO3 or aqua regia, to remove any trace of metal. Was any of this done, between runs or after a number of runs?
Response 4. Thank you for your kind comment. We utilized distilled water for cleaning according to the procedure described in previously reported works for the green synthesis of metal nanoparticles from plant extract [1,2]. That is why we did not use such type of acid/strong oxidizer during this synthesis process. Please see sections 2.2 and 2.3 of the revised manuscript for the details.
- Shaik MR, Khan M, Kuniyil M, Al-warthan A, Alkhathlan HZ, Siddiqui MRH, Shaik JP, Ahamed A, Mahmood A, Khan M, Adil SF (2018) Plant-Extract-Assisted Green Synthesis of Silver Nanoparticles Using Origanum Vulgare L . Extract and Their Microbicidal Activities. Sustainability. 10(4) : 913. https://doi.org/10.3390/su10040913
- Gomathi M, Prakasam A, Rajkumar PV (2019) Green Synthesis, Characterization and Antibacterial Activity of Silver Nanoparticles Using Amorphophallus Paeoniifolius Leaf Extract. J Clust Sci. 30 : 995–1001. https://doi.org/10.1007/s10876-019-01559-y.
Comment 5. (Figure 1): How many replicates were performed for each synthesis? Also, how reproducible was each synthesis? What was the variation in SPR peak shape and maximum?
Response 5. Thank you for your kind comment. We synthesized both AuNPs and AgNPs for five times following the same procedure that we mentioned in the Sections 2.2 and 2.3 of the manuscript. For each time of synthesis, the observed color of the synthesized nanoparticles solution was found consistent. The Uv-vis peak wavelength did not vary either. However, the peak intensity varied slightly for the five different syntheses.
Comment 6. 167, 187, 188, 191: citations errors
Response 6. Thank you for your kind comment. We have added proper references for the sentences in the mentioned lines of the comments. We have also included DOI address for the refences. Please see references [36], [5,6], [6] for the lines 167, 187, and 188 in the revised manuscript. We have rechecked and added appropriate references for all the statements in the revised manuscript.
Comment 7. 185: I disagree with the statement that the NPs were “spherical”. One could tell with a higher resolution TEM, but not with the SEM provided.
Response 7. Thank you for your kind comment. TEM is the best analytical tool for the shape and size determination. But we could not perform the TEM analysis owing to its unavailability in our region. However, we have included high magnification FESEM images with the revised manuscript and updated our description accordingly. Please see Figure 3 and Section 3.3 of the revised manuscript.
Comment 8. 190: SEM is not an in situ method. As such, when we observe NP aggregation, we do not know whether it resulted from SEM sample preparation, or whether the aggregation was already present in the reaction mixture. That is different from stating that the aggregation observed comes from the SEM preparation (we do not know, although the SPR peak shape reported earlier likely indicates low aggregation).
Response 8. Thank you for your kind comment. We regret for any misleading information presented in the manuscript. We don’t know exactly how this aggregation occurred. We have modified the writing and included the aggregation process based on the SPR peak response. Please see the section 2.3 of the revised manuscript for the details.
Comment 9. 203: The biomolecules observed with FTIR are present in the sample. However, we do not know whether they are adsorbed on the NP surface (if stabilizing agents) or are residues within the mixture. Also, if they acted as reducing agents, we would see a chemical change, and I am not convinced that this is the case here.
Response 9. Thank you for your kind comment. We are very sorry about any confusion caused by our presentation. Previous works that utilized plant extracts for synthesizing Ag and Au NPs reported stabilization of nanoparticles through these biomolecules from plant extract [1−3]. However, the drastic decrease in peak intensity might be directly related to the low stability and subsequently, aggregation of the synthesized NPs [2,3]. We have clarified our explanation in the revised manuscript. Please see Section 3.4 of the revised manuscript.
[3] F. Azarbani and S. Shiravand (2020) Green synthesis of silver nanoparticles by Ferulago macrocarpa flowers extract and their antibacterial, antifungal and toxic effects. Green Chemistry Letters and Reviews 13:41-49. https://doi.org/10.1080/17518253.2020.1726504
Comment 10. 244-251: Although there is some consensus as to how silver NPs act as antimicrobial agents (by releasing Ag+ ions), I would write as “it is believed”, “research has indicated” etc. instead of describing it as a known fact.
Response 10. Thank you for your kind comment. We have modified our manuscript according to reviewer suggestion.
Comment 11. 252-254: Comment above applies to ROS here.
Response 11. Thank you for your kind comment. We have modified our manuscript as the reviewer suggested.
Comment 12. 272- and l. 280-: This is a very interesting addition. Experimentally, sunlight increases the rate of nucleation and/or growth in the synthesis silver nanoparticles. The “proposed” mechanism as to how sunlight actually acts in silver NP formation is not a fully known fact, and should not be described as such.
Response 12. Thank you for your comment. We have changed the explanation of the effect of sunlight on the AgNP growth mechanism according to the reviewer’s suggestion. Please see Section 3.6 of the revised manuscript for the details.
Comment 13. 291-: Mechanisms are always “proposed” and can never be 100% “proven”. Here the authors describe a mechanism without any quantitative experimental data to support it. That is not acceptable. They can “propose” a mechanism based on their qualitative observations and from what Is known in the literature, as an interesting addition. That would be fundamentally different from advancing a mostly unfounded mechanism.
Response 13. Authors thank reviewer for pointing out this mistake. We have corrected it as a probable mechanism based on our experimental findings and literature review. Please see the section 3.7 of the revised manuscript.
Reviewer 2 Report
The work submitted for review concerns the production of typical nanoparticles using plant systems. Important factors in the development of nanotechnology are methods of obtaining nanomaterials using biological methods, which include processes using bacteria, fungi, and plant extracts. Together with
the development of nanotechnology, ecological methods have become an alternative to processes traditional ones, which do not always comply with the principles of green chemistry and are not indifferent to the natural environment. It is worth emphasizing the attempts to search for new solutions, although the selection of plant systems often becomes rather local.
Several aspects require additional clarification:
- The average crystallite size of the nanoparticles was calculated on pages 4 and 5. The authors calculate the average crystallite size as 41.06 nm and 26.6 nm for AuNP and AgNP respectively. In this case, the indication of which lattice planes such calculation was made. From only the first one or more?. Only them the correct size can be determined.
- There is no information on how to measure EDS. Was the measurement carried out from a point or an area?
- The authors suggest that "The peaks position and intensity of peaks of both AuNPs and AgNPs was changed, compared to the tuber extract (control) spectrum, indicated that, biomolecules present in tuber extract were responsible for the reduction and stabilization of synthesized AuNPs and AgNPs." This short and general explanation should be improved.
- English should be checked.
- The issue of the advisability of the proposed approach must be highlighted.
Author Response
2nd Reviewers' Comments:
The work submitted for review concerns the production of typical nanoparticles using plant systems. Important factors in the development of nanotechnology are methods of obtaining nanomaterials using biological methods, which include processes using bacteria, fungi, and plant extracts. Together with
the development of nanotechnology, ecological methods have become an alternative to processes traditional ones, which do not always comply with the principles of green chemistry and are not indifferent to the natural environment. It is worth emphasizing the attempts to search for new solutions, although the selection of plant systems often becomes rather local.
Comment 1. The average crystallite size of the nanoparticles was calculated on pages 4 and 5. The authors calculate the average crystallite size as 41.06 nm and 26.6 nm for AuNP and AgNP respectively. In this case, the indication of which lattice planes such calculation was made. From only the first one or more? Only them the correct size can be determined.
Response 1. Thank you for your kind comment. We have tabulated the value of crystallite sizes for all assigned peaks obtained from the respective XRD spectrum of AuNPs and AgNPs. Please see Table 1 in the revised manuscript. We used the assigned peaks of XRD data to calculate the average crystallite size for AuNP and AgNP and updated this information in the revised manuscript. Please see section 3.2 of the revised manuscript for details.
Comment 2. There is no information on how to measure EDS. Was the measurement carried out from a point or an area?
Response 2. Thank you for your kind comment. The EDS data was taken from specific points. We have mentioned this in the experimental section of the revised manuscript. Please see Section 2.4 of the revised manuscript.
Comment 3. The authors suggest that "The peaks position and intensity of peaks of both AuNPs and AgNPs was changed, compared to the tuber extract (control) spectrum, indicated that, biomolecules present in tuber extract were responsible for the reduction and stabilization of synthesized AuNPs and AgNPs." This short and general explanation should be improved.
Response 3. Thank you for your kind comment. We have thoroughly explained the effect of plant extract biomolecules on the FTIR spectrum as suggested. Please see the details in the sections 3.4 of the revised manuscript.
Comment 4. English should be checked.
Response 4. Thank you for your kind comment. We have revised the manuscript and improved the English writing as suggested. Please see the revised manuscript for the details.
Comment 5. The issue of the advisability of the proposed approach must be highlighted.
Response 5. Thank you for your kind comment. We have highlighted the advisability of the proposed approach in the introduction part as reviewer suggested. Please see the lines 59-68, page 2 of the revised manuscript.
Round 2
Reviewer 2 Report
The authors significantly improved the manuscript. It is now ready for publication.